# Optimization of Technological Parameters of Straw Fiber-Based Plant Fiber Seedling Pot Raw Materials

Ying Zhang, Qian-Jin Zhu, Shuai Gao, Shuang Liu, Long-Hai Li * and Hai-Tao Chen *

College of Engineering, Northeast Agricultural University, No. 600 Changjiang Road, Xiangfang District, Harbin 150030, China; zhangying83@neau.edu.cn (Y.Z.); 18434761319@163.com (Q.-J.Z.); g15581740103@163.com (S.G.); liushuang_0277@163.com (S.L.)
* Correspondence: lilonghai@neau.edu.cn (L.-H.L.); htchen@neau.edu.cn (H.-T.C.)

**Abstract:** Straw fiber seedling pots are a promising substitute for plastic seedling pots. The mixing mode of straw fiber affects the mechanical properties of the raw material membrane of the seedling pot. To explore the processing technology of making the raw material membrane of the seedling pot with two kinds of plant fibers in a layered manner, the optimal combination of the process parameters of the raw material membrane of the seedling pot without additives in the production process was studied experimentally. Response surface methodology (RSM) was used to analyze the parameters (beating degree of unbleached softwood kraft pulp fiber, beating degree of wheat straw fiber, wheat straw fiber quality percentage content, and film grammage) with regard to the dry tensile index and Z-direction tensile index of the seedling pot body. The optimal process parameter combination with a certain dry tensile index and Z-direction tensile index of seedling pot raw material was obtained by using four factors and five levels of a quadratic regression orthogonal rotation center combination design parameter optimization experiment. The optimal technical parameters were optimized as wheat straw fiber quality percentage content 70%, film grammage 70 g/m$^2$, unbleached softwood kraft pulp fiber beating degree 47–48 °SR, and wheat straw fiber beating degree 65–75 °SR. With the optimal conditions, the dry tensile index of the seedling bowl raw material film was between 21 and 22 N·(m·g$^{-1}$), and the Z-direction tensile index was greater than 2.1 N·(m·g$^{-1}$). Using wheat straw fibers and unbleached sulfite wood pulp fibers as raw materials for seedling pots, the raw material membrane of the seedling pots was made in a layered manner. The experimental study proved this feasibility. With this mixing process of raw materials, the straw fiber-based plant fiber seedling pot would meet the demands of a crop nursery after adding chemical additives. The research results provide a theoretical basis and technical support for the manufacture of the raw material membrane of the seedling pot body.

**Keywords:** central composite design; seedling pot; wheat straw; completely degradable; optimization



## 1. Introduction

Vegetable transplanting is an important link in the vegetable production process. Transplanting vegetables with seedling pots is an effective technical measure to increase production, quality, and efficiency, which can reduce the delayed period of seedlings in the field, increase the survival rate of seedlings, extend the growth period of seedlings, and increase the yield of vegetables. Raising vegetable seedlings in a nutrient pot can reduce the number of seeds used and cultivate strong seedlings. The standard seedling bowl made of traditional plastic material is not easy to degrade and will have a negative impact on the ecological environment. Under the global ecological environment with the implementation of the "plastic limit order" and the "dual carbon" goal, fully degradable materials can replace plastic materials and be applied to seedling pots, which is in line with the concept of green development.

The common degradable seedling pots include paper pots, grass pots, nutrient pots, and biomass seedling pots. Using crop straw to make seedling pots can not only reduce

straw burning and effectively use straw [1], but also can degrade the straw-based materials into carbon dioxide and water after being treated by microorganisms in the soil, so that the biomass can circulate naturally and promote the benign development of the ecological environment. In related studies, some researchers studied the use of different raw materials to make seedling pots [2–6]; Zhang X et al. [7] explored the preparation technology and parameter optimization of rice seedling plates; other researchers studied the degradability of the seedling pots made by different raw materials and processes [8–10]; still other researchers studied the different performances of different degradable seedling pots [11–13]; Zhou Yong et al. [14] studied the degradable seedling bowl in the cultivation of Pinellia group application in seedling cultivation; and Postemsky P.D. et al. [15] studied the recovery of residual substrate in the cultivation of Ganoderma lucidum mushroom and its application as a biodegradable horticultural seedling container. However, there are few reports on the use of plant fibers to prepare raw materials for thin-walled seedling pots.

In order to eliminate the environmental pollution caused by plastic seedling pots, this research took the straw fiber-based plant fiber prepared by our research group as the main research object, and the technology of using plant fiber instead of plastic to be used as the raw material of the seedling pot; namely, wheat straw fiber and wood pulp fiber were laminated and mixed in a layered state, so as to explore the optimal combination of process parameters, and provide a theoretical basis and technical support for making seedling pots with straw fiber-based plant fiber.

## 2. Materials and Methods

### 2.1. Materials and Equipment

The following materials were used in the study: Ji Mai 26-type wheat straw was harvested in Shandong province of China in June 2018. Unbleached softwood kraft pulp fiber was sold in the Chinese market.

The following equipment was used in the research: D200 straw fiber pretreatment machine (working temperature 25 to 120 °C, processing speed 0 to 140 r/min) developed by Northeast Agricultural University of China, Harbin, China; ZT4-00 Valley beater (Zhongtong Test Equipment Co., Ltd., Xingping, China); ZJG-100 Schopper's beating degree instrument (Changchun Yueming Scientific Instruments Co., Ltd., Changchun, China); ZCX-A manual sheet former (Changchun Yueming Scientific Instruments Co., Ltd., Changchun, China); DRK101A paper product quality detector (Jinan Nake Industry and Trade Co., Ltd., Jinan, China); JA5003B electronic balance (0.001 g precision Shanghai Jingke Tianmei Scientific Instruments Co., Ltd., Shanghai, China); and a straw-rubbing and -cutting machine (Harbin Longmu Machinery Company, Harbin, China).

### 2.2. Methods and Procedure

Based on the single factor pre-test, aiming at saving costs and improving efficiency, this experiment selected beating degree of unbleached softwood kraft pulp fiber, beating degree of wheat straw fiber, wheat straw fiber quality percentage content, and film grammage as experimental factors, the dry tensile index and Z-direction tensile index were used as performance indicators, and a four-factor and five-level quadratic regression orthogonal rotation center combination design test method was adopted. The factor level coding table is shown in Table 1.

The unbleached softwood kraft pulp fiber was beaten according to the laboratory beating Valley beater, as per GB/T24325 (2009) [16]; the wheat straw fiber was beaten according to the method in document [17], according to the beating degree of wheat straw fiber required in Table 1; the beating degree was measured according to Schopper–Riegler method, as per GB/T 3332 (2004) [18], according to the wheat straw fiber quality percentage content required in Table 1; and the film grammage was taken to prepare the hand-made sheet according to the fast cassette method, as per GB/T 24326 (2009) [19]. Different hand-made sheets were overlapped and dried. After being placed at 20 °C and 35 ± 5% relative humidity for 24 h, the dry tensile index was measured with reference to the tensile strength

determination method stated in GB/T 12914 (2008) [20], and the Z-direction tensile index was measured with reference to the method for measuring the bonding strength between paper and paperboard in literature [21]. Design-Expert software (version 6.0.10) was used to statistically analyze the data.

**Table 1.** Experimental factors coded by levels.

| Levels | Beating Degree of Unbleached Sulfate Conifer Pulp Fiber $X_1/^\circ$ SR | Pulpability of Wheat Straw Fiber $X_2/^\circ$ SR | Wheat Straw Fiber Quality Percentage Content $X_3/\%$ | Film Grammage $X_4/$ (g/m²) |
|---|---|---|---|---|
| 2 | 70 | 90 | 90 | 90 |
| 1 | 60 | 80 | 80 | 80 |
| 0 | 50 | 70 | 70 | 70 |
| −1 | 40 | 60 | 60 | 60 |
| −2 | 30 | 50 | 50 | 50 |

## 3. Results and Discussion

### 3.1. Experimental Data

Thirty-six experiments were conducted according to the processing parameters in Table 2, and the experimental results are listed in Table 2.

**Table 2.** Experimental plan and results.

| No. | Factors | | | | Response | |
|---|---|---|---|---|---|---|
| | Beating Degree of Unbleached Sulfate Conifer Pulp Fiber $X_1/^\circ$ SR | Beating Degree of Wheat Straw Fiber $X_2/^\circ$ SR | Wheat Straw Fiber Quality Percentage Content $X_3/\%$ | Film Grammage $X_4/$ (g/m²) | Dry Tensile Index $y_1/(\text{N}\cdot(\text{m}\cdot\text{g}^{-1}))$ | Z-Direction Tensile Index $y_2/(\text{N}\cdot(\text{m}\cdot\text{g}^{-1}))$ |
|---|---|---|---|---|---|---|
| 1 | 40 | 60 | 60 | 60 | 21.152 | 0.933 |
| 2 | 60 | 60 | 60 | 60 | 34.476 | 2.938 |
| 3 | 40 | 80 | 60 | 60 | 22.857 | 0.814 |
| 4 | 60 | 80 | 60 | 60 | 35.976 | 2.190 |
| 5 | 40 | 60 | 80 | 60 | 19.524 | 1.133 |
| 6 | 60 | 60 | 80 | 60 | 37.048 | 3.162 |
| 7 | 40 | 80 | 80 | 60 | 20.857 | 1.276 |
| 8 | 60 | 80 | 80 | 60 | 36.948 | 3.024 |
| 9 | 40 | 60 | 60 | 80 | 21.314 | 1.195 |
| 10 | 60 | 60 | 60 | 80 | 34.610 | 3.343 |
| 11 | 40 | 80 | 60 | 80 | 22.914 | 1.005 |
| 12 | 60 | 80 | 60 | 80 | 35.171 | 2.943 |
| 13 | 40 | 60 | 80 | 80 | 20.429 | 1.033 |
| 14 | 60 | 60 | 80 | 80 | 36.038 | 3.238 |
| 15 | 40 | 80 | 80 | 80 | 18.457 | 0.913 |
| 16 | 60 | 80 | 80 | 80 | 34.105 | 2.429 |
| 17 | 30 | 70 | 70 | 70 | 15.524 | 0.648 |
| 18 | 70 | 70 | 70 | 70 | 43.805 | 4.048 |
| 19 | 50 | 50 | 70 | 70 | 32.524 | 1.724 |
| 20 | 50 | 90 | 70 | 70 | 30.833 | 1.452 |
| 21 | 50 | 70 | 50 | 70 | 28.505 | 1.486 |
| 22 | 50 | 70 | 90 | 70 | 24.910 | 2.157 |
| 23 | 50 | 70 | 70 | 50 | 28.238 | 2.171 |
| 24 | 50 | 70 | 70 | 90 | 24.833 | 2.367 |
| 25 | 50 | 70 | 70 | 70 | 22.716 | 2.100 |
| 26 | 50 | 70 | 70 | 70 | 21.390 | 2.257 |
| 27 | 50 | 70 | 70 | 70 | 20.848 | 2.210 |
| 28 | 50 | 70 | 70 | 70 | 22.086 | 2.190 |
| 29 | 50 | 70 | 70 | 70 | 22.241 | 2.514 |
| 30 | 50 | 70 | 70 | 70 | 21.971 | 2.233 |
| 31 | 50 | 70 | 70 | 70 | 22.981 | 2.429 |
| 32 | 50 | 70 | 70 | 70 | 22.105 | 2.524 |
| 33 | 50 | 70 | 70 | 70 | 21.048 | 2.619 |
| 34 | 50 | 70 | 70 | 70 | 21.410 | 2.143 |
| 35 | 50 | 70 | 70 | 70 | 21.562 | 2.219 |
| 36 | 50 | 70 | 70 | 70 | 22.867 | 2.619 |

### 3.2. Influences of Factors on Dry Tensile Index

The variance analysis results of the dry tensile index are shown in Table 3. The results indicate that the quadratic regression model could be used to describe the influence of factors on the dry tensile index. The fitting equation was significant with the F-value 190.17 and the *p*-value less than 0.0001 under the confidence $\alpha$ = 0.05. The lack of fit with a *p*-value of 0.2387 was not significant relative to the pure error, indicating that the equation fits well. The quadratic regression models could predict dry tensile index accurately according to the model Equation (1):

$$
\begin{aligned}
y_1 = {}& 21.94 + 7.23X_1 - 0.03X_2 - 0.51X_3 - 0.53X_4 + 1.86X_1{}^2 + 2.37X_2{}^2 + 1.13X_3{}^2 \\
& + 1.08X_4{}^2 - 0.16X_1X_2 + 0.8X_1X_3 - 0.2X_1X_4 - 0.5X_2X_3 - 0.39X_2X_4 - 0.31X_3X_4
\end{aligned}
\tag{1}
$$

where $y_1$ represents the dry tensile index (N·m/g), and $X_1$, $X_2$, $X_3$, and $X_4$ indicate the beating degree of unbleached sulfate conifer pulp fiber (°SR), beating degree of wheat straw fiber (°SR), wheat straw fiber quality percentage content (%), and film grammage (g/m$^2$), respectively.

**Table 3.** Variance analysis of regression model of dry tensile index.

| Source | | Sum of Squares | DF | Mean Square | F Value | *p* Value |
|---|---|---|---|---|---|---|
| Dry tensile index | model | 1654.37 | 14 | 118.17 | 190.17 | <0.0001 ** |
| | $X_1$ | 1253.23 | 1 | 1253.23 | 2016.82 | <0.0001 ** |
| | $X_2$ | 0.02 | 1 | 0.02 | 0.03 | 0.8608 |
| | $X_3$ | 6.26 | 1 | 6.26 | 10.07 | 0.0046 ** |
| | $X_4$ | 6.63 | 1 | 6.63 | 10.66 | 0.0037 ** |
| | $X_1{}^2$ | 111.29 | 1 | 111.29 | 179.10 | <0.0001 |
| | $X_2{}^2$ | 179.51 | 1 | 179.51 | 288.88 | <0.0001 |
| | $X_3{}^2$ | 40.54 | 1 | 40.54 | 65.25 | <0.0001 |
| | $X_4{}^2$ | 37.51 | 1 | 37.51 | 60.37 | <0.0001 |
| | $X_1X_2$ | 0.43 | 1 | 0.43 | 0.70 | 0.4122 |
| | $X_1X_3$ | 10.36 | 1 | 10.36 | 16.68 | 0.0005 ** |
| | $X_1X_4$ | 0.66 | 1 | 0.66 | 1.06 | 0.3147 |
| | $X_2X_3$ | 4.04 | 1 | 4.04 | 6.50 | 0.0187 * |
| | $X_2X_4$ | 2.39 | 1 | 2.39 | 3.84 | 0.0634 |
| | $X_3X_4$ | 1.50 | 1 | 1.50 | 2.41 | 0.1355 |
| | Residual | 13.0491 | 21 | 0.62 | | |
| | Lack of Fit | 7.6490 | 10 | 0.76 | 1.56 | 0.2387 |
| | Pure Error | 5.4001 | 11 | 0.49 | | |
| | Cor Total | 1667.4151 | 35 | | | |

Notes: ** represents highly significant ($p < 0.01$), and * represents significant ($p < 0.05$).

By comparing the *p*-values, it was found that the beating degree of unbleached sulfate conifer pulp fiber, wheat straw fiber quality percentage content, and film grammage had the highest influence on the dry tensile index and the smallest *p*-values. This was followed by beating degree of unbleached sulfate conifer pulp fiber, film grammage, and wheat straw fiber quality percentage content. The interactive effect between the beating degree of unbleached sulfate conifer pulp fiber and the wheat straw fiber quality percentage content was highly significant, with a *p*-value of 0.0005. The interactive effect between beating degree of wheat straw fiber and wheat straw fiber quality percentage content was significant, as the *p*-value was 0.0187.

The 3D response surface between beating degree of unbleached sulfate conifer pulp fiber and wheat straw fiber quality percentage content on the dry tensile index is shown in Figure 1a. The dry tensile index increased with the increase in the beating degree of unbleached sulfate conifer pulp fiber. This was due to the increase in the degree of fiber splitting, which increased the bonding force between the fibers. Under the condition of constant quantitative, the dry tensile index was directly proportional to the bonding force between fibers, and the increase in the bonding force increased the dry tensile index. When the wheat straw fiber mass percentage content was high, the dry stretch index was low; this was because the mass of one fiber increases and the mass of the other fiber decreases

under a certain quantitative condition. As the binding force between the filaments of the unbleached sulfate conifer pulp fiber was stronger than that between the filaments of wheat straw fiber, the quality of the unbleached sulfate conifer pulp fiber decreased, which reduced the overall binding force of the fiber.

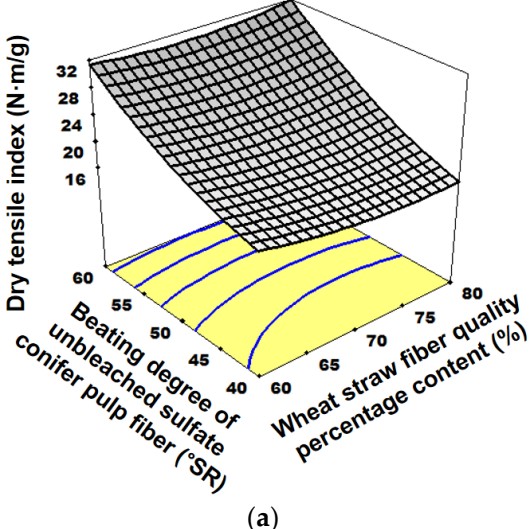
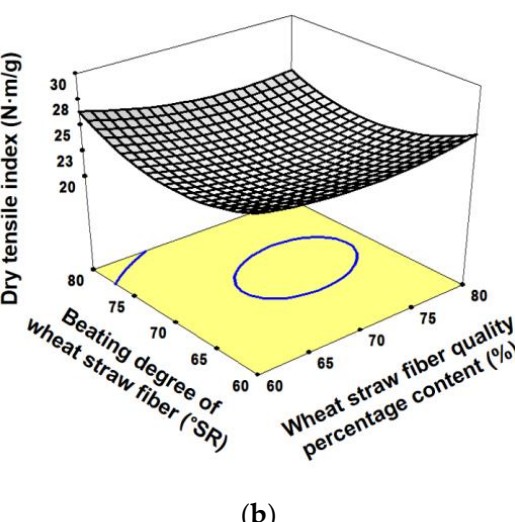

(**a**)　　　　　　　　　　　　　　　(**b**)

**Figure 1.** Response surface plot of factors affecting dry tensile index: (**a**) Influence of beating degree of unbleached sulfate conifer pulp fiber and wheat straw fiber quality percentage content on dry tensile index; (**b**) Influence of beating degree of wheat straw fiber and wheat straw fiber quality percentage content on dry tensile index.

The 3D response surface between beating degree of wheat straw fiber and wheat straw fiber quality percentage content on the dry tensile index is shown in Figure 1b. The dry tensile index decreased with the increase in wheat straw fiber quality percentage content, which was caused by the decrease in the content of unbleached sulfate conifer pulp fiber. The dry tensile index was lower when the beating degree of wheat straw fiber was at a medium level. The reason was that when the beating degree of wheat straw fiber was in the middle level, the properties of processed fiber were weakened, the number of fine fibers was increased, and the fiber bonding strength was reduced. With the further improvement in beating degree of wheat straw fiber, when beating degree of the wheat straw fiber was at a high level, the dry tensile index increased. The reason was that the processed fine fibers ran away with water through the filter screen at that time, and the aspect ratio of remaining fibers increased, which improved the overall bonding force of the fibers; or the number of fibers separated from the fiber bundles increased, and the degree of fiber splitting increased; or the fiber bundles were further cut off, so that the exposed number of fiber bundle cross-sections increased, and the number of fibers brushed at the cross-sections increased, which further increased the bonding force between the fibers.

### 3.3. Influences of Factors on Z-Direction Tensile Index

The variance analysis results for Z-direction tensile index (Table 4) indicate that the quadratic regression model could be used to describe the influence of factors on the Z-direction tensile index. Under the confidence level of $\alpha = 0.05$, the fitting equation was significant with an F-value of 47.62 and *p*-value less than 0.0001. The lack of fit with *p*-value 0.5660 was not significant relative to the pure error, which indicated that the equation fit well. The quadratic regression models could accurately predict the Z-direction tensile index according to the model Equation (2).

$$y_2 = 2.34 + 0.91X_1 - 0.12X_2 + 0.09X_3 + 0.04X_4 - 0.19X_2{}^2 - 0.13X_3{}^2 \\ -0.02X_4{}^2 - 0.11X_1X_2 + 0.04X_1X_4 + 0.03X_2X_3 - 0.04X_2X_4 - 0.16X_3X_4 \qquad (2)$$

**Table 4.** Variance analysis of regression model of Z-direction tensile index.

| Source | | Sum of Squares | DF | Mean Square | F Value | *p* Value |
|---|---|---|---|---|---|---|
| | Model | 22.82 | 14 | 1.63 | 47.62 | <0.0001 ** |
| | $X_1$ | 19.74 | 1 | 19.74 | 576.60 | <0.0001 ** |
| | $X_2$ | 0.36 | 1 | 0.36 | 10.42 | 0.0040 ** |
| | $X_3$ | 0.20 | 1 | 0.20 | 5.83 | 0.0249 * |
| | $X_4$ | 0.04 | 1 | 0.04 | 1.26 | 0.2742 |
| | $X_1{}^2$ | 0.00 | 1 | 0.00 | 0.01 | 0.9232 |
| | $X_2{}^2$ | 1.19 | 1 | 1.19 | 34.85 | <0.0001 ** |
| | $X_3{}^2$ | 0.58 | 1 | 0.58 | 16.97 | 0.0005 ** |
| Z-direction | $X_4{}^2$ | 0.02 | 1 | 0.02 | 0.49 | 0.4927 |
| tensile index | $X_1 X_2$ | 0.20 | 1 | 0.20 | 5.97 | 0.0235 * |
| | $X_1 X_3$ | 0.00 | 1 | 0.00 | 0.00 | 0.9680 |
| | $X_1 X_4$ | 0.03 | 1 | 0.03 | 0.77 | 0.3904 |
| | $X_2 X_3$ | 0.02 | 1 | 0.02 | 0.52 | 0.4802 |
| | $X_2 X_4$ | 0.03 | 1 | 0.03 | 0.79 | 0.3836 |
| | $X_3 X_4$ | 0.42 | 1 | 0.42 | 12.27 | 0.0021 ** |
| | Residual | 0.7188 | 21 | 0.03 | | |
| | Lack of Fit | 0.3223 | 10 | 0.03 | 0.89 | 0.5660 |
| | Pure Error | 0.3964 | 11 | 0.04 | | |
| | Cor Total | 23.5394 | 35 | | | |

Notes: ** represents highly significant ($p < 0.01$); and * represents significant ($p < 0.05$).

In Equation (2), $y_2$ represents the Z-direction tensile index (N·m/g), and $X_1$, $X_2$, $X_3$, and $X_4$ represent the beating degree of unbleached sulfate conifer pulp fiber (°SR), beating degree of wheat straw fiber (°SR), wheat straw fiber quality percentage content (%), and film grammage (g/m$^2$), respectively. By comparing the *p*-values, it was found that the beating degree of unbleached sulfate conifer pulp fiber and beating degree of wheat straw fiber had the highest influence on the Z-direction tensile index, with the smallest *p*-value. This was followed by the beating degree of unbleached sulfate conifer pulp fiber, the beating degree of wheat straw fiber, and the wheat straw fiber quality percentage content. The interactive effect between the beating degree of unbleached sulfate conifer pulp fiber and the beating degree of wheat straw fiber was significant, as the *p*-value was 0.0235. The interactive effect between the wheat straw fiber quality percentage content and the film grammage was highly significant, with a *p*-value of 0.0021.

The 3D interaction response graph of the beating degree of unbleached sulfate conifer pulp fiber and the beating degree of wheat straw fiber on the Z-direction tensile index is shown in Figure 2a. The Z-direction tensile index increased with the increasing beating degree of unbleached sulfate cone pulp fiber, which was due to the fact that in the beating degree within the experimental range, after wood pulp fiber was processed, the degree of fiber splitting broom increased with the increasing beating degree, and after fiber splitting broom, the bonding force between fibers increased, and both the same fiber and the dissimilar fiber increased, thus increasing the separation force of the fibers between different layers. When the beating degree of wheat straw fiber was at a medium level, the Z-direction tensile index was slightly higher than that at both ends. It was found that when the beating degree of wheat straw fiber was at a medium level, the fine fibers existed on the membrane surface and did not flow away with the water. The fiber splitting degree in this layer was high, which enhanced the fiber-bonding force between the layers, thus enhancing the layering force. When the amount was fixed, the tensile index in the Z-direction increased.

The interaction response surface for the Z-direction tensile index between the parameters of film grammage and the wheat straw fiber quality percentage content is shown in Figure 2b. When the content of wheat straw fiber quality percentage was low, the Z-direction tensile index increased with the increase in the film grammage. It was found that when the quantitative increased, the content of wood pulp fiber increased, and the number of fiber bonds between layers and the fiber separation force increased. When the wheat straw fiber quality percentage content was high, the Z-direction tensile index decreased with the increase in the film grammage. It was found that the fiber content of wheat straw increased with the quantitative increase, and the degree of splitting and

binding force of the wheat straw fiber was weaker than that of wood pulp fiber, and the binding force between the higher number of wheat straw fiber layers and the lesser wood pulp fiber layers was decreased.

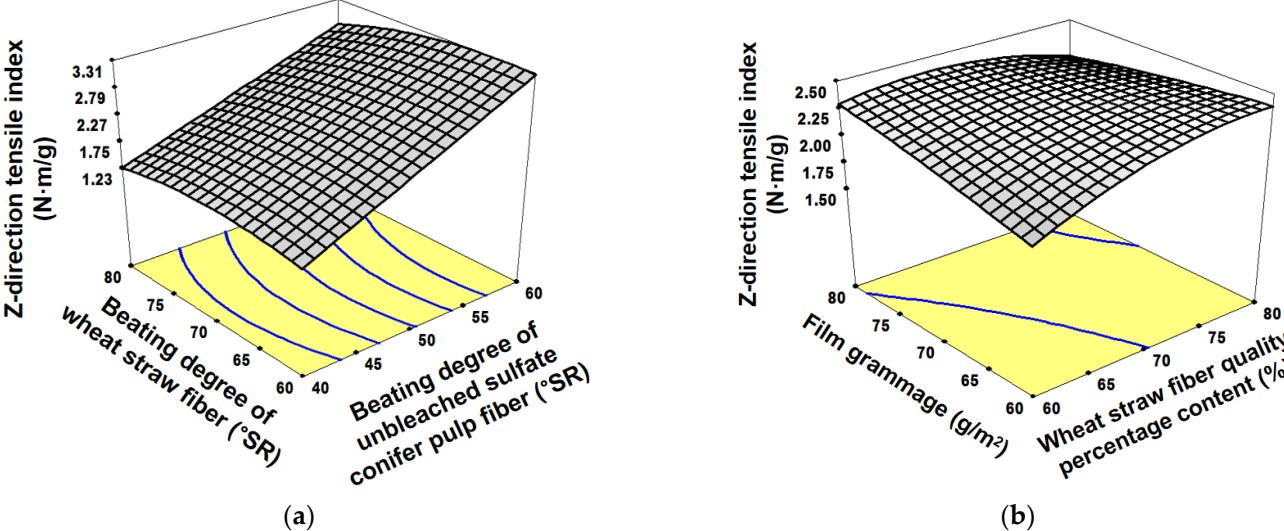

**Figure 2.** Response surface plot of factors affecting Z-direction tensile index: (**a**) Influence of beating degree of unbleached sulfate conifer pulp fiber and beating degree of wheat straw fiber on Z-direction tensile index; (**b**) Influence of wheat straw fiber quality percentage content and film grammage on Z-direction tensile index.

### 3.4. Optimization of Parameters

In order to meet the technical requirements for the production of straw fiber-based plant fiber seedling pot raw material film from wheat straw fiber, based on the principle of cost saving and efficiency improvement, the Design-Expert software was used to set the performance indicators, which required that the dry tensile index of seedling pot raw material film was between 21 and 22 N·(m·g$^{-1}$), the Z-direction tensile index was more than 2.1 N·(m·g$^{-1}$), and other factors were set as 0 level. The range of the beating degree of unbleached sulfate conifer pulp fiber and the beating degree of wheat straw fiber were optimized, as shown in Figure 3. The optimized combination of process parameters was as follows: wheat straw fiber content 70%, film grammage of seedling pot raw material 70 g/m$^2$, beating degree of unbleached sulfate conifer pulp fiber 47–48 °SR, and beating degree of wheat straw fiber 65–75 °SR, which met the target requirements.

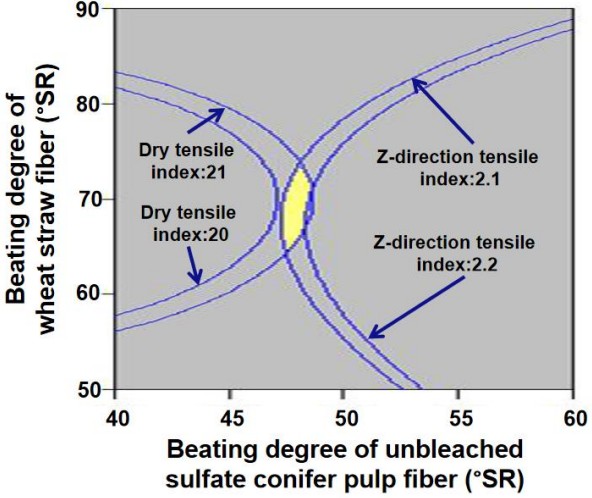

**Figure 3.** Optimum analysis of modulation adjustment parameters.

## 4. Conclusions

The process parameters of replacing the raw materials of plastic seedling bowls with plant fibers were studied. The results indicate that the beating degree of unbleached sulfate conifer pulp fiber and the wheat straw fiber content are the main factors influencing the performance of the seedling pot raw material film. The processing parameters of the wheat straw fiber-based plant fiber seedling pot raw materials were combined as follows: the beating degree of unbleached sulfate conifer pulp fiber 47–48 °SR, the beating degree of wheat straw fiber 65–75 °SR, wheat fiber content 70%, and the film grammage of seedling pot raw materials 70 g/m$^2$, which met the demands of the dry tensile index (between 21 and 22 N·m/g) and Z-direction tensile index of seedling pot raw material ($\geq$2.1 N·m/g). From the current research, it can be concluded from the experimental study that the raw materials of seedling pots made of wheat straw fiber and unbleached sulfite wood pulp fiber are feasible. In addition, the raw materials can meet the agronomic requirements of crop seedlings. The results provide a reference for the production and preparation of plant fiber-based seedling pots.

**Author Contributions:** Methodology, Y.Z. and H.-T.C.; software, Y.Z. and L.-H.L.; validation, Y.Z., Q.-J.Z., and S.G.; data curation, L.-H.L.; writing—original draft preparation, Y.Z.; writing—review and editing, L.-H.L.; project administration, S.L. This paper was prepared by the contributions of all authors. All authors have read and agreed to the published version of the manuscript.

**Funding:** This research was funded by the National Natural Science Foundation Youth Fund of China, grant number 31701311, Grant Number GA21B003, the Key Research and Development Projects in Heilongjiang Province in 2021, and the Science and Technology Project of Shandong Tobacco Monopoly Bureau (Company) in 2018.

**Institutional Review Board Statement:** Not applicable.

**Informed Consent Statement:** Not applicable.

**Data Availability Statement:** Not applicable.

**Conflicts of Interest:** The authors declare no conflict of interest.

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
