# Peer review of "Optimization of Technological Parameters of Straw Fiber-Based Plant Fiber Seedling Pot Raw Materials"

_applsci, doi:10.3390/app11157152_

Round 1

Reviewer 1 Report

The paper addresses an interesting topic for avoiding using plastic for seeding pot raw materials. The methods used seem correct.

Specific comments:

  • Line 84, something is missing in this phrase “And unbleached sulfate conifer pulp fiber.”
  • Include in the paper, the Coefficient of determination and the Coefficient of determination adjusted of the models indicated in Equation (1) and Equation (2).
  • Fig.3 must be changed, because the boxes inside the figure cover the lines.

Reviewer 2 Report

  • The conclusion is based on orthogonal experiment and response surface analysis but not the experimental mechanism analysis, therefore, it is suggested that the author should put forward the importance of this work from the perspective of practical engineering application. And these views should be reasonably presented in the abstract, introduction and discussion.
  • The second paragraph of the introduction should be further refined to show the motivation of the study.
  • The current English language level should be further improved.

Round 2

Reviewer 2 Report

Accept.